# Development of Intestinal Injury and Restoration of Weaned Piglets under Chronic Immune Stress

**DOI:** 10.3390/antiox11112215

**Published:** 2022-11-09

**Authors:** Jiayi Yu, Changbing Zheng, Jie Zheng, Geyan Duan, Qiuping Guo, Peiwen Zhang, Mengliao Wan, Yehui Duan

**Affiliations:** 1CAS Key Laboratory of Agro-Ecological Processes in Subtropical Region, Institute of Subtropical Agriculture, Chinese Academy of Sciences; Hunan Provincial Key Laboratory of Animal Nutritional Physiology and Metabolic Process, National Engineering Laboratory for Pollution Control and Waste Utilization in Livestock and Poultry Production, Scientific Observing and Experimental Station of Animal Nutrition and Feed Science in South-Central, Ministry of Agriculture, Changsha 410125, China; 2University of Chinese Academy of Sciences, Beijing 100039, China; 3College of Animal Science and Technology, Hunan Agricultural University, Changsha 410128, China

**Keywords:** weaned piglets, immune stress, intestinal injury and restoration

## Abstract

This study aimed to investigate the effects of lipopolysaccharide (LPS)-induced chronic immune stress on intestinal morphology and function, immune system, oxidative status, and mitochondrial function in piglets. Fifty healthy Duroc × Landrace × Yorkshire piglets (21 ± 2 days old, barrow, 6.98 ± 0.14 kg body weight) were selected and randomly allotted to five groups, which were slaughtered at 0 (0 group), 1, 5, 9, and 15 d of LPS injection. The results showed that compared with the piglets without LPS injection, LPS injection significantly impaired the intestinal morphology and permeability at 1, 5, and 9 d, as manifested by the increased serum lactic acid and decreased ratio of villus height to crypt depth (*p* < 0.05). Moreover, intestinal inflammation and oxidative and mitochondrial injury were caused at 1 d, as manifested by upregulated IL-6 mRNA expression, increased malondialdehyde content, and impaired mitochondrial morphology (*p* < 0.05). However, these parameters were restored to levels identical to 0 group at 9~15 d, accompanied by significantly increased antioxidant capacity, enhanced protein expression of CD3+ and CD68+, and upregulated mRNA abundance of genes related to mitochondrial biogenesis and functions (*p* < 0.05). Collectively, these results suggest that the intestinal injury of piglets caused by chronic immune stress could be self-repaired.

## 1. Introduction

Pigs experience biological stresses, such as environmental, physiological, and social challenges when weaned from their sow. Immune stress at one of the most stressful events, the weaning of piglets, can induce dysfunctions of intestine and immune system, which can subsequently lead to reduced feed intake as well as impaired pig growth and health [1]. Over recent years, improvements in nutrition and health have been used to ameliorate immune-stress-induced adverse effects [2,3,4,5,6,7]. However, our understanding of these improvements is incomplete for piglets, because in the actual pig farm environment, chronic rather than acute immune stress is common, and whereas the literature in this field is mainly based on the model of acute immune stress and intestinal injury which is established by injecting *Escherichia coli* lipopolysaccharide (LPS) once [4,6,8,9]. Therefore, a greater understanding of the dynamic effects of chronic immune stress on intestine barrier function is needed to improve strategies to overcome chronic immune stress and to improve the intestinal health of piglets. 

The synthesis and utilization of ATP are specifically required in the proliferation of intestinal epithelial cells to sustain intestinal integrity, intestinal function (including nutrient digestion and absorption), and the mitochondrial activity and metabolism to contribute to this process [10,11,12]. Apart from its central role of energy generation described above, the mitochondrial respiratory chain also produces intracellular reactive oxygen species (ROS), a predominant byproduct of electron transfer [13]. Accumulating and emerging lines of evidence have revealed a strong association between ROS and impaired intestinal function [14,15,16,17,18]. Specifically, large amounts of ROS can damage proteins, DNA, and lipids in the membrane components, leading to mitochondrial dysfunction and ultimately exacerbating intestinal damage [17,19,20]. Accordingly, to mitigate the prooxidant consequences of ROS, cells have developed complex system of antioxidant enzymes such as superoxide dismutase (SOD), glutathione peroxidases (GSH-Px), and myeloperoxidase (MPO) [21]. Additionally, ROS induced by mitochondria is thought to be a key factor in the stimulation of inflammation and the secretion of pro-inflammatory cytokines such as tumor necrosis factor (TNF-α), interleukin (IL)-6, and IL-1β [22]. To date, little is known regarding the effect of chronic immune stress on mitochondrial function, inflammation, and the antioxidative system in the intestine of piglets. Therefore, further investigation is warranted.

The immune stress model induced by a single injection of a certain dose of LPS exhibited drawbacks such as short duration and susceptibility to immune tolerance. However, previous studies have found that repeated administrations of incremental doses of LPS-induced chronic immune stress model can result in a systemic inflammation and damage to muscle, spleen, liver, and other tissues and organs, which weakens immune tolerance and significantly enhanced the immune stress response [23,24,25]. In particular, as for the intestine as a primary target during an endotoxic attack, it has been documented that LPS induced various impairments of intestinal morphological alterations, enhancements of mucosal permeability and bacterial translocation [23,26]. Therefore, the present experiment was carried out to investigate the effects of chronic immune stress on intestinal barrier function, mitochondrial function, immune defense, and oxidative status by establishing a chronic immune stress model through alternate day incremental injection of LPS to mimic the effects of adverse stimuli on immune stress in weaned piglets in the actual pig farm environment. 

According to our knowledge, this is the first study to reveal the development and modulation of intestinal injury and restoration in weaned piglets on a longitudinal timeline, thus providing a new perspective in the use of nutrients to shorten the duration of damage caused by immune stress in piglets and blazing a new ground for the reduction of production costs.

## 2. Materials and Methods

### 2.1. Animals and Diets

The experiment was performed according to the Chinese guidelines for animal welfare and experimental protocols, and all procedures involving animal subjects were approved by the Animal Care Committee of Institute of Subtropical Agriculture, Chinese Academy of Sciences (ISA-2020-0005).

Fifty healthy Duroc × Landrace × Yorkshire piglets (6.98 ± 0.14 kg, and barrow) were selected and weaned at 21 d of age. After a 7 d adaptation, they were randomly allotted to 5 groups with 10 replicate per treatment. In a temperature-controlled nursery barn (25 to 27 °C), the piglets were kept separately in 1.80 × 1.10 m cages with unrestricted access to clean water and food. The piglets in the control group (group 0 d) were injected with 0.90% sterile saline and slaughtered using electrical stunning at 1 d before the trial. From day 1 of the trial, the piglets in the other four groups were challenged with LPS (*E. coli* serotype 055: B5; Sigma Chemical, St Louis, MO, USA), which was dissolved in sterile saline with the initial dose of 80 μg per kg body weight, and then increased by 30% every other day [25], and slaughtered at 1, 5, 9, and 15 d of the trial (groups 1 d, 5 d, 9 d, and 15 d). Piglets were weighed individually at the start and final day and feed consumption was recorded. The scheme of the experimental design was shown in Figure 1. All piglets were provided the same diets during the feeding experiment and the ingredients and nutritional values are shown in Table 1, which met the National Research Council (2012) nutrient requirement [27]. The experimental period lasted for 15 days.

### 2.2. Sample Collection

On days 1, 5, 9, 15 of the trial, blood samples were obtained by venipuncture through the anterior vena cava, then centrifuged for 15 min at 3000× *g* at 4 °C, and the supernatant was stored in a refrigerator at −80 °C for serum biochemistry. All piglets were stunned with an electrical stunner set at 250 V and 0.5 A with a current flow for 6 s following blood collection at 3 h. Immediately after slaughter, the ileum and jejunum were emptied of contents by rinsing with normal saline. The middle small intestine tissue samples (3-cm sections) were collected and then respectively fixed in 4% paraformaldehyde solution and 2.5% glutaraldehyde solution for analysis of intestinal morphology [10], and gently inverted twice. A piece of the small intestine the size of a grain of rice was put in the electron microscope solution pending analysis. The contents of the small intestine were rinsed with sterile saline after being intercepted with sterile medical scissors for around 10 cm. After that, the intestine section was cut along the midline longitudinally, and the inner wall of the small intestine was gently scraped with a sterilized slide at 45 degrees angle to the tissue, and the scraped mucosa was wrapped with tin foil and rapidly frozen in liquid nitrogen and then stored at −80 °C used for subsequent real-time PCR analysis.

### 2.3. Analysis of Serum Biochemical Indexes 

The plasma samples were thawed out of the −80°C refrigerator and centrifuged for 10 min at 800× *g* at 4 °C, then 150 μL of supernatant was aspirated in a serum glass, which was detected using Automatic Biochemical Analyzer (Roche, Basel, Switzerland) and commercial kits (Sino-German Beijing Leadman Biotech Ltd., Beijing, China). The concentrations of blood indicators were detected included the following: diamine oxidase (DAO) and lactic acid (LACT).

### 2.4. Intestinal Morphology

Approximately 3–4 μm thick microtome sections of intestinal tissues were obtained from each sample and stained with hematoxylin and eosin (H&E) as previously described [28]. The hematoxylin-eosin stained images of intestine were captured and measured with computer-assisted microscopy (Micrometrics TM; Nikon ECLIPSE E200, Tokyo, Japan). The apparent characteristics of microvillus image and the villus height and crypt depth were observed and measured using CaseViewer software.

### 2.5. Redox Parameters Detection

The grinded intestinal mucosal was analyzed for the detection of malondialdehyde (MDA), superoxide dismutase (SOD), glutathione peroxidases (GSH-Px), and total antioxidant capacity (T-AOC), according to the manufacturer’s instructions (Nanjing Jiancheng Bioengineering Institute, Jiangsu, China) specific for porcine (MDA assay kit/A003-4-1, SOD assay kit/A001-3-1, GSH assay kit/A061-1-1, T-AOC assay kit/A015-2-1). Results are shown by enzyme activity.

### 2.6. Quantitative Real-Time PCR

The extraction of total jejunal and ileal RNA was carried out with Trizol reagent (Invitrogen, Carlsbad, CA, USA) and the concentration of RNA was measured using NanoDrop^®^ ND-1000 (Thermo Fisher, Wilmington, DE, USA). Then, reverse transcription was performed according to the instructions of the TaKara reverse transcription kit (Takara Bio Inc., Tokyo, Japanese) as previously studied [29]. RT-PCR primer design was performed using Oligo 6.0 (Molecular Biology Insights, Inc.) software design with β-actin as the endogenous reference gene, which is listed in Table 2. Polymerase chain reaction was performed using LightCycler^®^ 480II quantitative PCR instrument (384 wells) (Roche, Basel, Switzerland) according to the guidelines of the SYBR Green Assay Kit (TaKaRa), with the following procedure: incubation for 30 s at 95 °C, followed by 40 cycles of denaturation for 95 °C × 5 s, annealing and extension for 60 °C × 30 s. The relative mRNA expression level of the gene of interest was calculated using the 2^−ΔΔCt^ method.

### 2.7. Immunofluorescence

The fixed fresh small intestinal tissues were dehydrated using dehydration series (70% ethanol, 96% ethanol, 100% ethanol), followed by removal of the dehydrated reagents using xylene three times for 30 min each. The cassettes with constructs were quickly transferred to beakers with melted paraffin wax and left to rest in the oven for 1–2 days (60 °C), and the pre-treated tissue blocks were embedded with melted paraffin wax in specific embedding molds to form tissue wax blocks. Intestinal tissue paraffin sections were dewaxed in absolute ethanol, then placed them in a repair box filled with EDTA antigen recovery solution (pH 8.0) which was prepared in a microwave oven [28]. Subsequently, the sections were added, with BSA incubating for 30 min and then the primary antibody was added overnight at 4°C. Next, the sections were incubated with Alexa Fluor^®^ 488-conjugated or Alexa Fluor^®^ 594-conjugated second antibody (1:400) for 1 h at 37 °C [30]. After the sections were slightly dried, the nuclei were stained with DAPI and incubated at room temperature for 10 min. Then the sections were placed on the confocal laser scanning microscopy (Olympus, Tokyo, Japan) to capture the images. The data of LPS treatments was normalized to 0 d group. Primary antibodies against Drp1 were purchased from Cell Signalling Technology (Danvers, MA, USA). Anti-8-OHdG was purchased from Bioss (Woburn, MA, USA). Anti-CD3, and CD68 were purchased from AiFang biological (Changsha, China). 

### 2.8. Mitochondrial Ultrastructure

Fixed tissue blocks were composed within the electron microscopy fixation fluid through the process of trimming, post-ES fixation, dehydration, resin infiltration, etc. Imported epoxy resin 812 was used as the embedding agent, and the tissue specimen was embedded in the resin. Next, a Slicing Machine (Leica, Wetzlar, Germany) was used to slice resin and stain with uranium acetate and lead citrate. Then the dyed copper mesh was observed on the Transmission Electron Microscope (Hitachi, Ltd., Tokyo, Japan) which captured the reasonable view. 

### 2.9. Statistical Analysis

The individual piglet used in this experiment served as the experimental unit for all data collection. One-way analysis of variance (ANOVA) was used for experiments with more than two groups by SAS 8.2 software (Institute, Inc., Cary, NC, USA) followed by Duncan’s multiple comparisons. The average ± standard error (Mean ± SEM) was used to express the results of the data. The nonparametric Kruskal-Wallis test was used when the data with non-normal distributions or homogeneity, which was presented in the boxplot as median with interquartile range. Differences were considered statistically significant at *p* < 0.05. 

## 3. Results

### 3.1. The Intestinal Morphology and Function under LPS-Induced Chronic Immune Stress

As revealed in Table 3, serum DAO activity was the lowest at 15 d of LPS injection and the highest at 1, 5, and 9 d, with the intermediate value at 0 d (*p* < 0.05). Serum LACT level was the highest at 1 d and was the lowest at 0 and 15 d, with intermediate values at 5 and 9 d (*p* < 0.05).

In jejunum (Figure 2B), compared with the value at 0 d, the villus height was significantly decreased at 1 d, and was restored to the similar value at 5, 9, and 15 d as that at 0 d (*p* < 0.05). Compared with the 0 d, the crypt depth was significantly increased at 1, 5, 9, and 15 d (*p* < 0.05), but no differences were detected between the four groups (*p* > 0.05). Compared to the 0 d, the ratio of villus height to crypt depth (V/C) was significantly decreased at 1, 5, and 9 d, and was restored to the similar value at 15 d as that at 0 d (*p* < 0.05). In ileum (Figure 2B), the villus height at 1 d was of similar value at 0 d, then significantly decreased at 5 and 9 d, and was restored to the similar value at 15 d as that at 0 d (*p* < 0.05). The crypt depth was the highest at 1 d and the lowest at 0 and 15 d, with intermediate values at 5 and 9 d (*p* < 0.05). Compared to the 0 d, the V/C ratio was significantly decreased at 1, 5, and 9 d, and was restored to the similar value at 15 d as that at 0 d (*p* < 0.05).

The relative mRNA expression of tight junction protein (TJP) genes was presented in Figure 2C. In jejunum, the ZO1 mRNA expression was not significantly different among the groups (*p* > 0.05). The OCLN mRNA expression was the highest at 9 d, followed by 5 d, with the lowest values observed 0, 1, and 15 d (*p* < 0.05), and no difference was observed among the three groups (*p* > 0.05). In ileum, compared to the 0 d, the relative mRNA expression of ZO1 and OCLN tended to reduce at 1 d, albeit not statistically significant (*p* > 0.05). There was no significant difference in the mRNA expression of ZO1 and OCLN between 0 d and the other groups (*p* > 0.05).

### 3.2. Intestinal Epithelial Inflammation under LPS-Induced Chronic Immune Stress

In jejunum (Figure 3A), the quantification of CD3+-positive cells reached the highest at 9 d (*p* < 0.05), whereas no differences were obtained among the rest of four groups (*p* > 0.05). Compared with the 0 d, the quantification of CD68+-positive cells significantly increased at 5, 9 and 15 d and showed an increased trend at 1 d (*p* > 0.05). In ileum (Figure 3B), the quantification of CD3+ and CD68+-positive cells both achieved the highest at 9 d (*p* < 0.05).

The effects of chronic immune stress on mRNA expression of pro-inflammatory genes (TNF-α and IL-6) and anti-inflammatory cytokine IL-4 in the intestine were examined. In jejunum (Figure 3C), there was no significant differences observed in mRNA relative expression of NF-κB among all the groups (*p* > 0.05), nonetheless, significant increases of IL-6 mRNA relative expression were observed at 1 d compared with other groups (*p* < 0.05) and restored to similar value at 15 d as that at 0 d (*p* < 0.05). With the comparation of 0 d, the relative mRNA expression of IL-4 significantly declined at 5, 9 and 15 d (*p* < 0.05). In ileum (Figure 3C), compared with 0 d, significant upregulation of NF-κB gene was measured at 15 d and then pro-inflammatory gene expression of TNF-α was also significantly increased at 15 d (*p* < 0.05). Significant increases of relative IL-6 mRNA expression were observed at 1 d compared with other groups (*p* < 0.05). Under LPS-induced chronic immune stress, the relative mRNA expression of anti-inflammatory cytokines IL-4 significantly elevated caused by LPS at 5 d, 9 d, 15 d compared with 0 d (*p* < 0.05).

### 3.3. The MDA and Anti-Oxidative Parameters under LPS-Induced Chronic Immune Stress

In jejunum, as presented in Figure 4A, the content of MDA was highest at 1 d. In comparison to 0 d, T-AOC was significantly declined at other days. No significant differences were detected for SOD activities. The activity of GSH-Px was the lowest at 15 d (*p* < 0.05) and showed a declined trend at 5 and 9 d, albeit not of statistical significance (*p* > 0.05). In ileum (Figure 4A), the MDA content came up to the highest at 1 d and lowest at 5 d and 9 d (*p* < 0.05). Compared with 0 d, the T-AOC was significantly reduced at 5 d and 9 d (*p* < 0.05). In addition, an increase in the activities of SOD at 9 d and 15 d were observed compared to the 0 d (*p* < 0.05). But there no differences were observed in GSH-Px (*p* > 0.05). 

Real time-qPCR was performed to determine the mRNA expression of anti-oxidative related genes. In jejunum (Figure 4B), the decrease in the HMOX1 mRNA expression was observed at 5 d and 9 d in comparison to of 0 d (*p* < 0.05). No difference was observed in relative mRNA expression of SOD1 and Mn-SOD among the groups (*p* > 0.05). In ileum (Figure 4B), the relative mRNA expression of SOD1 was remarkably elevated at 15 d compared to that in the other groups (*p* < 0.05). The relative mRNA expression of HMOX1 achieved its highest mark at 5 d and then reverted to a similar value at 15 d as that at 0 d (*p* < 0.05). In comparation with 0 d, the relative mRNA expression of Mn-SOD significantly increased at 1 d, 5 d, and 15 d (*p* < 0.05). 

### 3.4. Intestinal Mitochondria Function and DNA Damage under LPS-Induced Chronic Immune Stress

As revealed in Figure 5A, a complete inner membrane and wider cristae in ileal mitochondria structure were observed at 0 d and 15 d, and the mitochondrial structural alteration including mitochondria swelling, scanty cristae, and vacuolization were observed on other days. However, structural alteration to jejunal mitochondria persisted until 15 d.

In jejunum, as shown in Figure 5B, the expression of dynamin-related protein 1 (Drp1) significantly increased at 9 and 15 d and 8-hydroxy-2′-deoxyguanosine (8-OHdG) reached the highest mark from 5 d to 9 d (*p* < 0.05). In ileum (Figure 5C), the protein expression of Drp1 was significantly increased at 5 and 9 d compared to those at 0 d, and reverted to a value similar to 0 d at 15 d (*p* < 0.05). Additionally, an increase in protein expression of 8-OHdG was observed with a maximum level at 1 d (*p* < 0.05).

The mRNA levels of several mitochondrial biogenesis and functions related genes were assessed to investigate the effect of chronic immune stress on mitochondrial function. In jejunum, as the results performed in Figure 6, the relative mRNA expression levels of PGC-1α were clearly upregulated at 5 d, 9 d and 15 d compared with 0 d (*p* < 0.05). The TFAM mRNA expression was significantly upregulated at 5 d and 9 d, while restored to a similar value at 15 d as that at 0 d (*p* < 0.05). A notable downregulation in the mt-SSB mRNA expression was observed at 9 d and 15 d compared to 0 d (*p* < 0.05). Chronic immune stress had no effect on the AMPK mRNA expression (*p* > 0.05) among all the groups. The relative mRNA expressions of glucokinase were significantly lower at 9 d and 15 d than that observed in the 0 d group. Chronic immune stress downregulated the relative mRNA expression of CS at 1 d but reverted to a similar value at 0 d, 5 d, 9 d and 15 d (*p* < 0.05). Compared with the 0 d, chronic immune stress reduced the expression of CcOX IV at 5 d and 9 d and increased the expression of CcOX V at 9 d, while both reverted to a value similar to 0 d at 15 d. Cyt c and NRF-1 mRNA expression were downregulated at 5 d and 15 d compared with 0 d (*p* < 0.05). Meanwhile, chronic immune stress reduced the ATPS mRNA expression at 5 d while increasing it at 15 d (*p* < 0.05). In ileum, chronic immune stress significantly triggered the upregulation of PGC-1α mRNA expression at 5 d, 9 d and 15 d (*p* < 0.05), albeit no differences were observed between 0 d and 1 d (*p* > 0.05). The AMPK mRNA expression achieved its highest point at 15 d with intermediate values at 1 d, 5 d and 9 d (*p* < 0.05). A significant increase of mt-SSB mRNA expression was observed at 15 d compared with other four groups (*p* < 0.05). The CS mRNA expression was the highest at 15 d and lowest at 1 d with intermediate value at 0 d, 1 d, and 9 d (*p* < 0.05). Meanwhile, the mRNA expression of CcOX I was increased at 15 d and no differences were observed in measurements from the other days (*p* < 0.05). A reduction in mRNA expression of CcOX IV was observed at 9 d compared to other days (*p* < 0.05). The relative mRNA of CcOX V achieved the highest value at 15 d and lowest value at 0 d with intermediate values at 1 d, 5 d and 9 d (*p* < 0.05). Compared with 0 d, the relative mRNA expression of Cyt c was significantly upregulated at 15 d (*p* < 0.05). Additionally, the mRNA expression of sirt1, TFAM, glucokinase, ATPS and NRF-1 were not influenced by chronic immune stress (*p* > 0.05).

## 4. Discussion

The small intestine mucosa play a vital role in feed-derived nutrient absorption and digestion, but also serve as the first defensive line that prevents intestinal epithelium invasion by pathogens of a hostile environment [31,32]. Many papers indicate that piglets reared in commercial facilities are exposed to continuous immune stimulations, andcan develop intestinal dysfunction [6,8,33,34]. Moreover, the increased intestinal permeability and intestine barrier dysfunction correlate with the translocation of bacteria from the intestine lumen into the blood circulation [35,36,37]. In the present study, the permeability of intestinal mucous membrane was assessed by the serum DAO and D-lactate. The DAO, an endocellular enzyme, which is present in high concentrations in the intestinal mucosa and synthesized by intestinal epithelial cells, can permeate from the impaired intestinal mucosa into the blood circulation [38]. D-lactate is produced by intestinal flora inherent fermentation; ordinarily the concentration of D-lactate in the blood is extremely low due to the metabolism of methylglyoxal. However, when the intestinal mucosal barrier is impaired and intestinal permeability is increased, D-lactate in the intestine enters the blood circulation, resulting in elevated blood levels of D-lactate [39,40]. Here, we found that LPS injection elevated the serum D-lactate level in the early to mid-term (1~9 d) but that it was restored to a similar value at 15 d as that at 0 d. Throughout the experiment, serum DAO activity was unaffected by LPS injection but it reached its highest level in the first to middle period (1 to 9 d). In line with these findings, intestinal morphology is compromised in the early period (1~5 d) in response to an LPS challenge and restored to normal at 15 d after LPS injection. The data herein indicated that the intestinal villus height shortened, the crypt depth deepened and the turnover rate of intestinal epithelial cells decreased in the early stages (1~5 d) of chronic immune stress, while the intestinal architecture was slowly repaired in the later stages and was not significantly different from the 0 d by 15 d. 

Moreover, with respect to the effect of chronic immune stress on intestinal architecture, the mRNA expression of tight junction proteins could further elucidate the mechanism of the impaired intestinal barrier integrity in weaned piglets at the molecular level. TJP allows the boundary of each adjacent epithelial cell to remain in close contact, sealing the intestinal surface and acting as a selective filter so that the intestine can efficiently absorb nutrients [41,42]. The mRNA abundance of both ZO1 and OCLN in ileum were lowest in the early stage (1 d) and highest in the middle and late stages (9~15 d), indicating that chronic immune stress impaired the mucosal layer of the intestinal tract in the early stage. These results were consistent with the results of intestinal morphology and permeability. Overall, the results suggest that chronic immune stress had a significant effect on the intestinal mucosal permeability and structure of the piglets, as evidenced by increased intestinal permeability and structural damage in the early stages of chronic immune stress, while the intestine may have potential to repair itself in the later stages.

Considering the correlation between immune function and intestinal mucosal injury and repair, we next analyzed immune function-related parameters [43,44]. Typically, T-cells and macrophages exert important roles in immune function (Sjödahl et al., 2014). Most peripheral T cells exist as resting lymphocytes until they receive an activation signal, which is regulated by the signal transmission of T cell receptor TCR/CD3 and soluble mediators such as cytokines and chemokines [45]. Notably, CD3 antigen is unique to T cells and its antibody is taken as an effective marker for lymphocyte analysis in routine experiments [46]. Macrophages are key players in the inflammatory responses of innate immune system due to their highly plastic phenotypes and great diversity [47,48]. CD68, a transmembrane glycoprotein, is highly expressed in the surface of macrophages [49,50]. Therefore, we analyzed intestinal activities of CD3+ (T-cell marker) and CD68+ (the macrophage-specific marker) to evaluate immune function during the period of chronic immune stress. The current studies revealed that CD3+- and CD68+-positive proteins were significantly expressed at day 9, but were restored to normal by post (15 d), which is consistent with the alteration trend of intestinal morphology. These findings suggest that chronic immune stress-induced injury and repair of intestinal morphology are regulated by the immune system. Further evidence was obtained from the results of macrophages, T cells and enterocytes-secreted IL-6 (a pro-inflammatory cytokine) [51]. In contrast to the alteration trend of CD3 and CD68, the jejunoileal mRNA expression of IL-6 was significantly elevated in the first phase (1 d), but was rapidly downregulated and returned to the levels obtained at 0 d in the following days (5~15 d). IL-6 can alter intestinal mucosal permeability by affecting epithelial cell TJP [52]. Therefore, it is speculated that in response to chronic immune stress, the restoration of the impaired intestinal morphology was associated with the improved immune function and downregulated mRNA expression of IL-6.

To determine the oxidative injury caused by chronic immune stress, the levels of MDA, 8-OHdG, SOD, GSH-PX, T-AOC, and MPO in jejunum and ileum of piglets were studied. The degree of lipid oxidative injury can be directly reflected by the level of MDA, which is the most common product of lipid peroxidation [16]. It has been reported that during DNA replication, 8-OHdG can be formed from abundant oxidative DNA lesion [53,54]. In the current study, lipid peroxidation in jejunum and ileum mainly appeared at 1 d of after LPS injection, and then significantly ameliorated, as manifested by the results of MDA activity and 8-OHdG expression in jejunum and ileum. The complex system of antioxidant systems (such as SOD) plays key roles in guarding the organism against the detrimental prooxidants [14]. SOD serves to convert O^2−^ to hydrogen peroxide (H_2_O_2_), thus scavenging O^2−^ [55]. In this study, we found that ileal SOD activity gradually increased from 0 to 15 d after LPS injection, and ileal T-AOC activity decreased at 1, 5, and 9 d after LPS injection and then recovered at 15 d. Moreover, the ileal SOD activity curve followed a time course that was consistent with the mRNA expression of SOD-related genes (SOD1 and Mn-SOD). However, the SOD activity in jejunum did not change significantly under the condition of chronic immune stress, which was consistent with the results of the mRNA expression of SOD-related genes (SOD1 and Mn-SOD). SOD1, also known as copper/zinc superoxide dismutase (Cu/Zn-SOD), can catalyze the conversion of superoxide to oxygen and H_2_O_2_ and endow stability to protein structure [56]. Mn-SOD, a critical antioxidant enzyme in mitochondria, can protect against oxidative damage by converting ROS to H_2_O_2_ and oxygen [57]. These findings suggest that lipid oxidative injury can be ameliorated at 15 d after LPS injection in both jejunum and ileum, and the improved antioxidant capacity might only be responsible for the diminished lipid oxidative injury in ileum rather than in jejunum.

To further clarify the effects of chronic immune stress on mitochondrial function in jejunum and ileum of piglets, we investigated the mitochondrial morphology by utilizing a transmission electron microscope and conducted the expression of mitochondrial function-related proteins and genes. It has been observed that chronic immune stress dramatically distorted and fragmented the mitochondrial structure of jejunum and ileum at 5–9 d after LPS injection, while the mitochondria reverted to their original and intact morphology at 15 d after LPS injection. Drp1, a dynamin-related protein in mitochondria, is critical for the regulation of mitochondrial fission [58]. But to date, the information regarding the Drp1 expression on intestinal injury induced by the immune stress of piglets is less well understood. Therefore, our study was conducted to detect the protein expression of Drp1 in the intestine using immunofluorescence, and the results showed a significant increase in its protein expression at 5 d and 9 d of LPS injection and a restoration to the 0 d level in late stages (15 d). In combination with the above mitochondrial morphological results, we deduced that the excessive mitochondrial division brought on by Drp1 hyperfunction may result in mitochondrial fragmentation and structural and functional disorders at 5–9 d after LPS injection. PGC-1α, AMPK, and TFAM are genes encompassed in mitochondrial biogenesis [59]. The present study found that mitochondrial biogenesis in intestine was markedly boosted since the 5 d of LPS injection, as manifested by upregulated mRNA expression of PGC-1 and TFAM in jejunum and upregulated mRNA expression of PGC-1α and AMPK in ileum. The mitochondrial network is dynamic and is involved in regulating many aspects of mitochondrial biology including glycometabolism, ATP generation, tricarboxylic acid (TCA) cycle, electron-transport respiratory chain and other signaling [60]. Therefore, the mitochondrial dysfunction could dramatically result in the altering cellular and tissue homeostasis. In this study, chronic immune stress downregulated the jejunal mRNA abundance of CS coding for mitochondrial glucose metabolism at 1 d but reverted to a similar value to 0 d at 5, 9, and 15 d, and the ileal CS and ND4 mRNA abundances were both upregulated at 15 d after LPS injection. Notably, the mRNA abundance of electron-transport respiratory chain-related genes (CcOX IV, CcOX V, Cyt c, and ATPS) presented in a tissue-dependent manner and followed a time course, manifested as upregulated mRNA expression of jejunal CcOX V and Cyt c at 9 d of LPS injection and upregulated mRNA expression of ileal CcOX I, CcOX IV, CcOX V, and Cyt c at 15 d of LPS injection. Overall, the intestinal mitochondrial morphology and function were impaired in the first and middle stages and mitigated at late stages, and the increased mitochondrial biogenesis capacity may only be responsible for the improved mitochondrial function in ileum rather than in jejunum.

In conclusion, our findings demonstrate that the intestine injury and restoration of weaned piglets exhibits a dynamic pattern of change in response to chronic immune stress. The intestine showed significant inflammatory and oxidative levels, decreased mitochondrial function and impaired intestinal mucosal barrier in the early to mid-term, while the later stages (15 d) showed a restoration to day 0 levels of intestinal mucosal barrier, inflammatory response, antioxidant levels and partial mitochondrial activity, indicating that weaned piglets underwent self-repair mechanisms in the intestine. There may be some possible limitations in this study. Along the lines of our current study on the regulation of the immune system, antioxidant system and mitochondrial function on the piglet intestine barrier, there is still much to learn about how these systems interact. In future, it will still be necessary to further investigate the modulation of intestinal injury and restoration in piglets by supplemental nutrition.

## Figures and Tables

**Figure 1 antioxidants-11-02215-f001:**
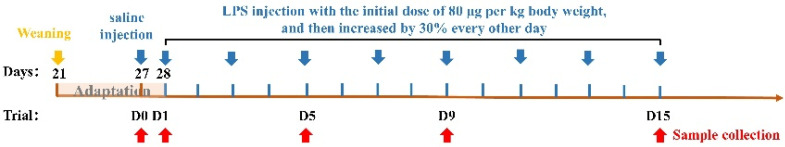
Effects Scheme of the experimental design. After a 7 d adaptation period, fifty piglets were randomly assigned to one of the five groups: (1) control group (pigs were injected with 0.90% sterile saline and slaughtered before 1 d of the trial); (2) LPS 1 d group (pigs were injected with LPS and slaughtered at 1 d of the trial); (3) LPS 5 d group (pigs were injected with LPS and slaughtered at 5 d of the trial); (4) LPS 9 d group (pigs were injected with LPS and slaughtered at 9 d of the trial); (5) LPS 15 d group (pigs were injected with LPS and slaughtered at 15 d of the trial). LPS, lipopolysaccharide.

**Figure 2 antioxidants-11-02215-f002:**
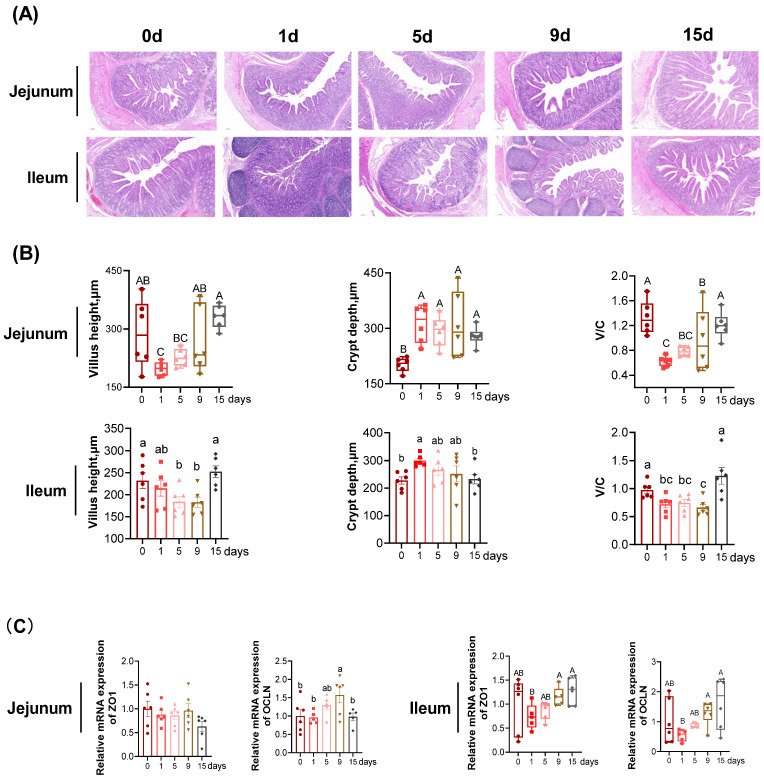
The intestinal mucosa barrier function under LPS-induced chronic immune stress: (**A**) representative hematoxylin-eosin stained image of intestine (magnification 10×); (**B**) quantification analysis of intestinal villus height, crypt depth, and ratio of villus height to crypt depth (V/C); (**C**) gene expression of tight junction proteins in intestinal mucosa. The two end edges of the box correspond to inter quartile ranges of data, the inside line represents the median, and the colored dot, squared, regular triangle, inverted triangle and diamond block in the figure correspond to the outliers for different days. n = 6 for each group. ^a,b,c^ Values within a row with different lowercase letters differ significantly (*p* < 0.05) by the Duncan’s multiple comparisons. ^A,B,C^ Values within a row with different capital letters differ significantly (*p* < 0.05) by the Kruskal–Wallis test. ZO1, zonula occludens-1; OCLN, occluding; CLON1, claudin-1.

**Figure 3 antioxidants-11-02215-f003:**
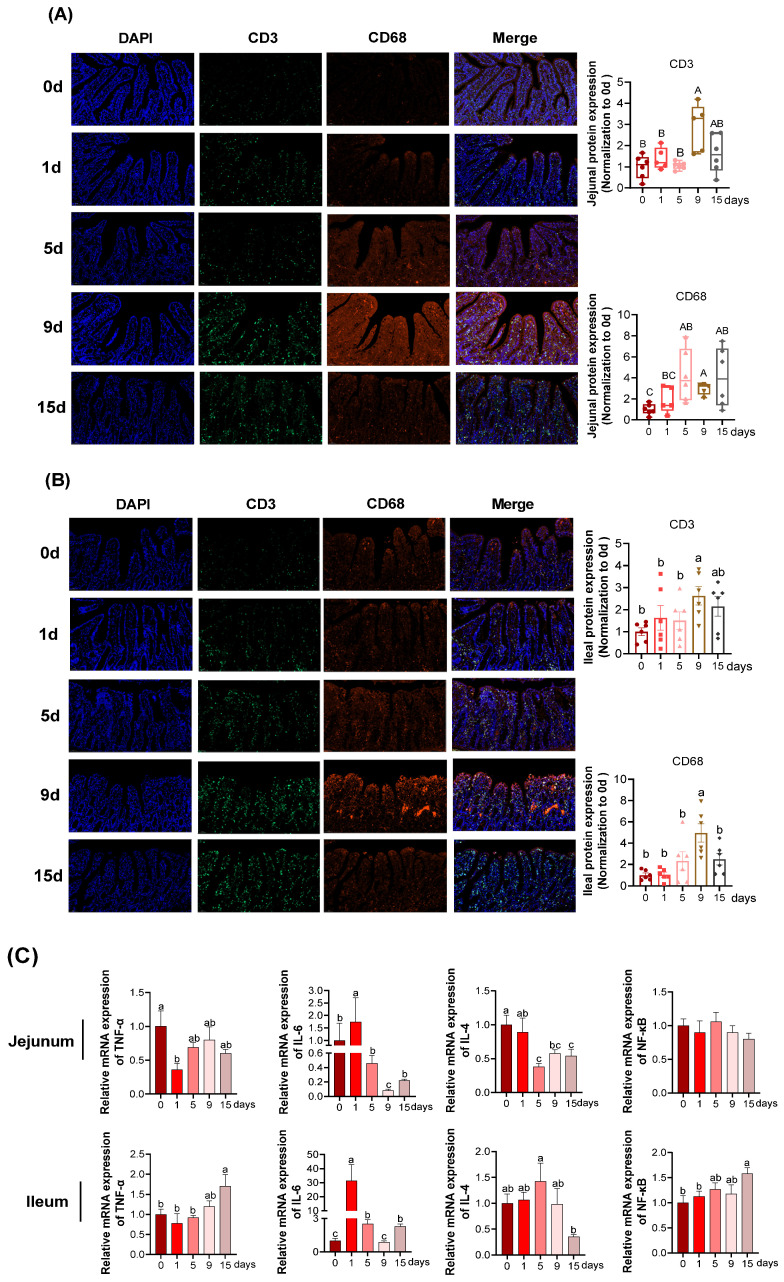
Intestinal epithelial inflammation under LPS-induced chronic immune stress: (**A**,**B**) representative images and quantification of CD3+ and CD68+-positive cells (A: jejunum; B: ileum); (**C**) the mRNA expression of inflammation related genes. The two end edges of the box correspond to inter quartile ranges of data, inside line represents the median, and the colored dot, squared, regular triangle, inverted triangle and diamond block in the figure correspond to the outliers for different days. n = 6 for each group. ^a,b,c^ Values within a row with different lowercase letters differ significantly (*p* < 0.05) by the Duncan’s multiple comparisons. ^A,B,C^ Values within a row with different capital letters differ significantly (*p* < 0.05) by the Kruskal–Wallis test. TNF-α, tumor necrosis factor-α; IL-6, interleukin-6; NF-κB, nuclear factor kappa-B; IL-4, interleukin-4.

**Figure 4 antioxidants-11-02215-f004:**
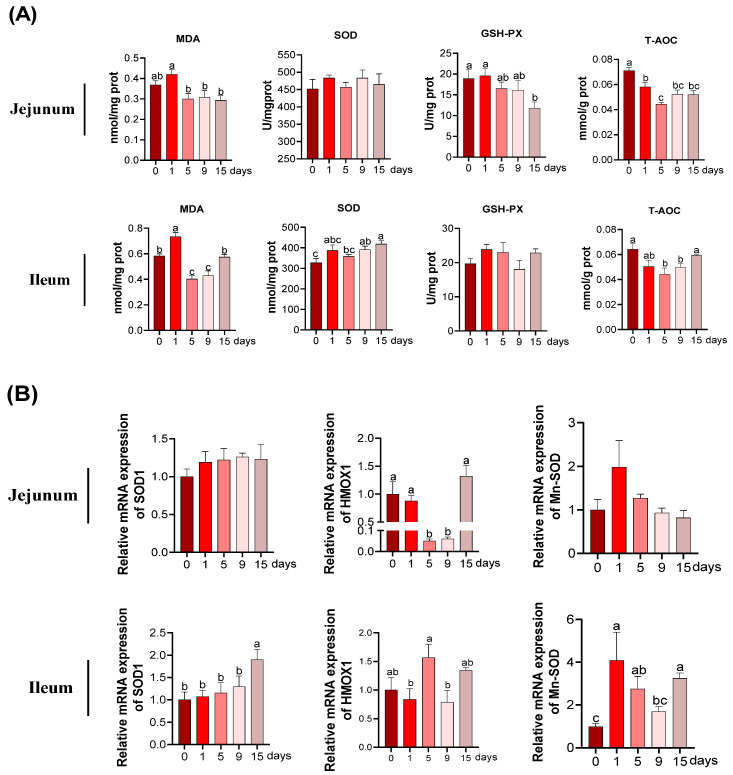
The MDA and anti-oxidative parameters under LPS-induced chronic immune stress: (**A**) the antioxidant enzymes activities and MDA content in jejunum and ileum of piglets; (**B**) the mRNA expression of anti-oxidative related genes. Data represents the mean ± SEM. n = 6. ^a,b,c^ Mean values with different letters were considered to be significantly different (*p* < 0.05). MDA, malondialdehyde; SOD, superoxide dismutase; GSH-Px, glutathione peroxidases; T-AOC, total antioxidant capacity; SOD1, copper and zinc superoxide dismutase; HMOX1, heme oxygenase 1; Mn-SOD, manganese-containing superoxide dismutase.

**Figure 5 antioxidants-11-02215-f005:**
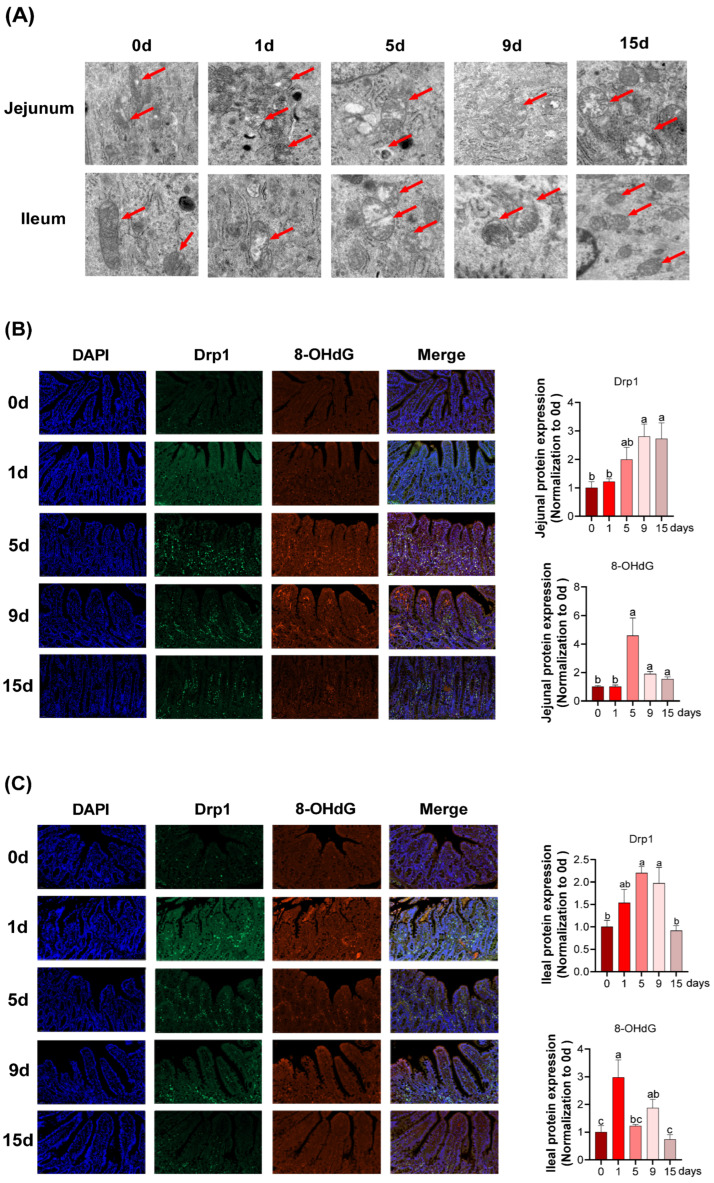
Intestinal mitochondria morphology and mitochondrial division-related proteins and DNA damage protein expresssion under LPS-induced chronic immune stress: (**A**) representative transmission electron microscopy images of intestinal mitochondria and the arrows in the figure point to the mitochondria; (**B**,**C**) representative images and quantification of Drp1 and 8-OHdG-positive proteins ((**A**): jejunum; (**B**): ileum). Data represents the mean ± SEM. n = 6. ^a,b,c^ Mean values with different letters were considered to be significantly different (*p* < 0.05).Drp1, dynamin-related protein 1; 8-OHdG, 8-hydroxy-2′-deoxyguanosine.

**Figure 6 antioxidants-11-02215-f006:**
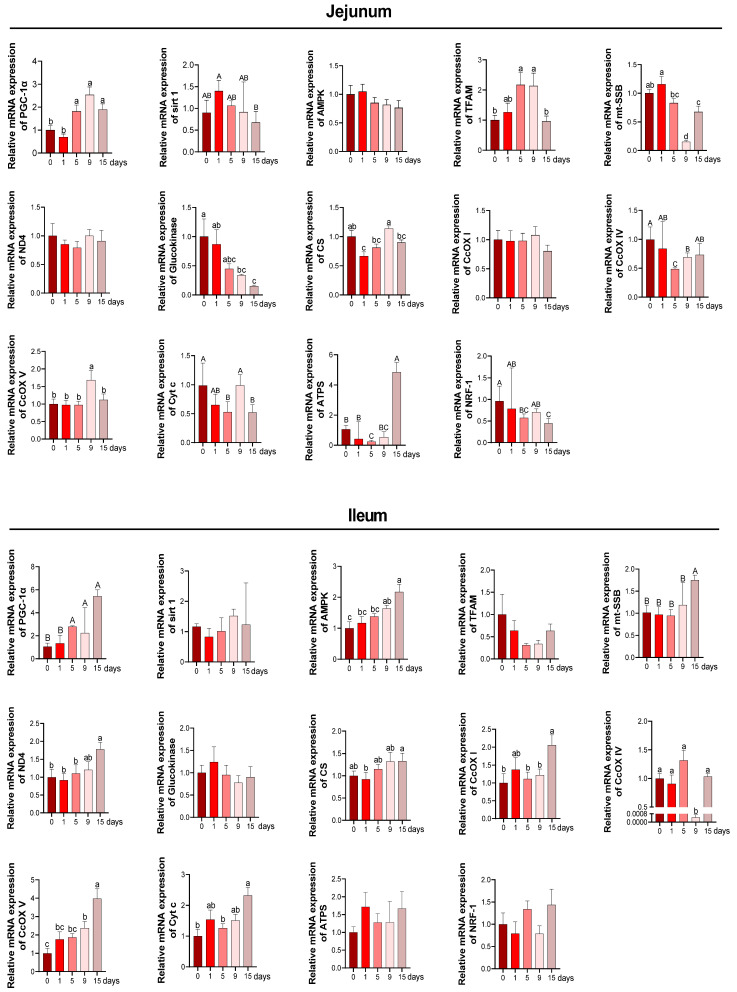
The mRNA expression of mitochondrial biogenesis and functions for related genes. n = 6 for each group. ^a,b,c^ Values within a row with different lowercase letters differ significantly (*p* < 0.05) by the Duncan’s multiple comparisons. ^A,B,C^ Values within a row with different capital letters differ significantly (*p* < 0.05) by the Kruskal–Wallis test. PGC-1α, peroxisome proliferator-activated receptor-g coactivator-1α; SIRT1, silent information regulator transcript 1; AMPK, AMP-activated protein kinase; TFAM, transcription factor a mitochondrial; mt-SSB, mitochondrial single-strand DNA-binding protein; ND4, NADH dehydrogenase subunit 4; CS, citrate synthase; CcOX I, cytochrome c oxidase I; CcOX IV, cytochrome c oxidase IV; CcOX V, cytochrome c oxidase V; Cyt c, Cytochrome c; ATPS, ATP synthase; NRF-1, nuclear respiratory factor-1.

**Table 1 antioxidants-11-02215-t001:** Composition and nutrient levels of the diets for weaned piglets (air-dried basis,%).

Items	Fresh Fish Oil
Ingredients, %	Contents, %
corn	30.14
Extruded corn	30.00
Soybean meal	9.00
Fish meal	7.00
Plasma protein powder	5.00
Whey powder	9.00
Glucose	3.00
Soybean oil	3.80
Limestone	1.05
Choline chloride	0.10
Antioxidants	0.05
Citric acid	0.50
Salt	0.10
Vitamin premix ^a^	0.30
Mineral premix ^b^	0.15
Lys 98%	0.45
DL-Met	0.20
L-Thr	0.14
L-Trp	0.02
Total	100.00
Nutrient content (%)	
DE (MJ/kg)	14.18
CP	18.61
Ca	0.80
Total P	0.56
Available P	0.38
Lys	1.37
Met + Cys	0.75
Thr	0.81
Trp	0.22

^a^ Vitamin premix supplied per kilogram of feed: 2200 IU Vitamin VA, 220 IU Vitamin D3, 0.5 mg Vitamin K3, 17.5 μg Vitamin B12, 3.5 mg riboflavin, 30 mg niacin, 10 mg D-pantothenic acid, 0.05 mg biotin, 0.3 mg folic acid, 1.0 mg thiamine, 7 mg pyridoxine, and 4.0 mg ethoxyquin. ^b^ Mineral premix supplied per kilogram of feed: 150 mg Fe (FeSO_4_), 100 mg Zn (ZnSO_4_), 30 mg Mn (MnSO_4_), 25 mg Cu (CuSO_4_), 0.5 mg I (KIO_3_), 0.3 mg Co (CoSO_4_), and 0.3 mg Se (Na_2_SeO_3_).

**Table 2 antioxidants-11-02215-t002:** Primers used for real-time PCR.

Gene	Sequence	Size (bp)	GeneBank No.
ZO1	F:ATCTCGGAAAAGTGCCAGGA R: CCCCTCAGAAACCCATACCA	172	XM_003480423.3
OCLN	F:CATGGCTGCCTTCTGCTTCATTGC R:ACCATCACACCCAGGATAGCACTCA	129	NM_001163647.2
CLDN1	F:TATGACCCCATGACCCCAGT R:GCAGCAAAGTAGGGCACCTC	108	NM_001244539.1
SOD1	F: GGTCCTCACTTCAATCCTGAATCC R: CACACCATCTTTGCCAGCAGT	102	NM_0011190422
HMOX1	F: CGCTCCCGAATGAACACTCT R: GCGAGGGTCTCTGGTCCTTA	148	NM_001004027.1
Mn-SOD	F: GGACAAATCTGAGCCCTAACG R: CCTTGTTGAAACCGAGCC	159	NM_214127
IL-4	F: GCTGCCCCAGAGAACACGAC R: AGGTTCCTGTCAAGTCCGCTC	119	NM_214123.1
TNF-α	F: AACCTCAGATAAGCCCGTCG R: ACCACCAGCTGGTTGTCTTT	129	NM_214022.1
NF-κB	F: GACAACATCTCCTTGGCGGG R: TCTGCTCCTGCTGCTTTGAGG	146	NM_001048232
PGC-1α	F: CCCGAAACAGTAGCAGAGACAAG R:CTGGGGTCAGAGGAAGAGATAAAG	111	NM 213963
SIRT1	F: TGACTGTGAAGCTGTACGAGGAG R: TGGCTCTATGAAACTGCTCTGG	143	EU030283.2
AMPK	F:ACCAGGACCCTTTGGCAGTT R:GAATCAGGTGGGCTTGTTGC	100	NM_001167633.1
TFAM	F: GGTCCATCACAGGTAAAGCTGAA R: ATAAGATCGTTTCGCCCAACTTC	167	AY923074.1
mt-SSB	F: CTTTGAGGTAGTGCTGTGTCG R: CTCACCCCTGACGATGAAGAC	143	AK352341.1
ND4	F: TTATTGGTGCCGGAGGTACTG R: CCCAGTTTATTCCAGGGTTCTG	112	NM 001097468
Glucokinase	F:CTTTTCCCTCCCACACTGCTAT R: GACTCCTCTTCCTGAGACCCTCT	119	AK233298.1
CS	F: CCTTTCAGACCCCTACTTGTCCT R: CACATCTTTGCCGACTTCCTTC	127	M21197.1
CcOX I	F: ATTATCCTGACGCATACACAGCA R: GCAGATACTTCTCGTTTTGATGC	127	AJ950517.1
CcOX IV	F: CCAAGTGGGACTACGACAAGAAC R: CCTGCTCGTTTATTAGCACTGG	131	AK233334.1
CcOX V Cyt c	F: ATCTGGAGGTGGTGTTCCTACTG R: GTTGGTGATGGAGGGGACTAAA F: TAGAAAAGGGAGGCAAACACAAG R: GGATTCTCCAGGTACTCCATCAG	160 154	AY786556.1 NM 001129970.1
ATPS	F: TGTCCTCCTCCCTATCACACATT R: TAGTGGTTATGACGTTGGCTTGA	116	AK230503
NRF-1	F: GCCAGTGAGATGAAGAGAAACG R: CTACAGCAGGGACCAAAGTTCAC	166	AK237171.1
β-actin	F: TCTTTTCCAGCCTTCCTTCTTG R: GAGGTCTTTACGGATGTCAACG	100	NM_007393

ZO1, zonula occludens-1; OCLN, occluding; CLDN1, claudin-1; SOD1, copper and zinc superoxide dismutase; HMOX1, heme oxygenase 1; Mn-SOD, manganese-containing superoxide dismutase; IL-4, interleukin-4; TNF-α, tumor necrosis factor-α; NF-κB, nuclear factor kappa-B; PGC-1α, peroxisome proliferator-activated receptor-g coactivator-1α; SIRT1, silent information regulator transcript 1; AMPK, AMP-activated protein kinase; TFAM, transcription factor a mitochondrial; mt-SSB, mitochondrial single-strand DNA-binding protein; ND4, NADH dehydrogenase subunit 4; CS, citrate synthase; CcOX I, cytochrome c oxidase I; CcOX IV, cytochrome c oxidase IV; CcOX V, cytochrome c oxidase V; Cyt c, Cytochrome c; ATPS, ATP synthase; NRF-1, nuclear respiratory factor-1.

**Table 3 antioxidants-11-02215-t003:** Serum biochemical indices under LPS-induced chronic immune stress.

Items	LPS Treatments					SEM	*p*-Value
	0 d	1 d	5 d	9 d	15 d		
DAO (mmol/L)	1.12 ^ab^	1.86 ^a^	1.92 ^a^	1.94 ^a^	0.74 ^b^	0.17	<0.05
LACT (mmol/L)	6.19 ^c^	12.49 ^a^	8.19 ^b^	8.03 ^b^	7.43 ^bc^	0.04	<0.05

DAO, diamine oxidase; LACT, lactate. ^a,b,c^ Values (n = 8) within a row with different superscripts differ significantly (*p* < 0.05).

## Data Availability

Not applicable.

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
