# Peer review of "Development of Intestinal Injury and Restoration of Weaned Piglets under Chronic Immune Stress"

_antioxidants, 2022, doi:10.3390/antiox11112215_

Round 1

Reviewer 1 Report

Citation of references in the text does not seem to be in line with the journal's requirements. Please check this.

A little more information about previous studies on immune stress model using LPS and about the influence of LPS on the intestine is needed in the introduction.

The novelty of study should be underlined in the end of the introduction

figures are too small and completely unreadable (especially micrographs). I suggest to redraft figures

What are the limitations of the experiment. Please present them in the discussion.

Author Response

Reviewer: 1

Comments to the Author

  1. Citation of references in the text does not seem to be in line with the journal's requirements. Please check this.

A: Thanks for the reviewer’s suggestion. Citation of references in the text has been improved according to the Journal’s requirements.

  1. A little more information about previous studies on immune stress model using LPS and about the influence of LPS on the intestine is needed in the introduction.

A: We sincerely appreciate the valuable comments. We have checked the paper carefully and added more information on immune stress model using LPS and the influence of LPS on the intestine into the INTRODUCTION part in the revised manuscript. (Line 70-79)

  1. The novelty of study should be underlined in the end of the introduction.

A: Considering the reviewer’s suggestion, we have added the novelty of this study in the end of INTRODUCTION. (Line 85-89)

  1. Figures are too small and completely unreadable (especially micrographs). I suggest to redraft figures

A: Thank you for pointing out this problem in our manuscript. According to the revised content, we have redrafted the figures to clearly show the experimental results.

  1. What are the limitations of the experiment. Please present them in the discussion

A: Thanks for the reviewer’s suggestion. The limitations of the experiments are now included in the revised manuscript. (Line 490-495)

Reviewer 2 Report

Title: Development of Intestinal Injury and Restoration of Weaned Piglets under Chronic Immune Stress

The paper reported the effect effects of lipopolysaccharide- induced chronic stress on intestinal morphology immune system, oxidative status, and mitochondrial function in piglets. The topic is interesting, actual and in line with journal objectives. The paper is well written, however, there are some points that should be clarified. I retain that the paper can be published in Antioxidants after minor revision.

Minor comments are reported below:

Line 39-42: The improvement in nutrition strategies is not only from a technological point of view. delete “technological”

Line 44: substitute” the vast majority of the published studies” with “the main literature”

Line 75-77: delete the sentence. It is redundant.

Line 84: Why the authors decided to employ barrows?

Line 86: How were the animals divided? One piglet per box? Add details.

Line 112-114: Delete the sentence that is a duplication.

Line 121: Specify the sampling site.

Line 129-135: rewrite for clarity.

Line 137: Substitute with “Microtome sections of intestinal tissues were obtained from each sample and stained…” Add the thickness of the sections.

Line 144: Substitute with: “The grinded intestinal mucosa was analyzed”

 Line 169-177: substitute slice with sections

Line 185 Statistical Analysis: Please add the experimental unit in the  statistical evaluation.

Line 214: correct “trended” in “tended”

Line 345: substitute” accumulating lines of literature” with “Many papers”

References: References in the text and in the references section are not in line with journal instruction for authors. Please revise this section.

Author Response

Reviewer: 2

Minor comments are reported below:

  1. Line 39-42: The improvement in nutrition strategies is not only from a technological point of view. delete “technological”

A: Thanks for the reviewer’s encouragement and suggestions. “technological” has been deleted in revised version.

  1. Line 44: substitute “the vast majority of the published studies” with “the main literature”

A: “the vast majority of the published studies” has been substituted with “the main literature”. (Line 44)

  1. Line 75-77: delete the sentence. It is redundant.

A: The sentence has been deleted in revised version.

  1. Line 84: Why the authors decided to employ barrows?

A: Firstly, the development of gonads exerts a big effect on the phenotype of pigs (Lubritz et al., 1991), and the detection of some markers (such as serum biochemistry index) may be hampered by excessive individual variability in androgen levels. Thus, the influence of gender on the experimental results should be avoided in animal experiments. Secondly, in the published articles on the porcine intestine, we observed that also using barrows as experimental materials (Liu et al., 2019; Pi et al., 2014; Wang et al., 2015). Thirdly, in the actual pig production farms, the barrows will be chosen to facilitate management and reduce feeding costs. Therefore, in this experiment, in order to remove the interference brought by gender and to better simulate the actual pig farm environment, the 21-day-old barrows were selected to investigate the developmental changes of intestinal tract under chronic immune stress.

References:

Lubritz, D., Johnson, B., and Robison, O.W. Genetic parameters for testosterone production in boars. J. Anim. Sci. 1991. 69, 3220-3224. 10.2527/1991.6983220x.

Liu, J., Zhang, Y., Li, Y., Yan, H., and Zhang, H. L-tryptophan Enhances Intestinal Integrity in Diquat-Challenged Piglets Associated with Improvement of Redox Status and Mitochondrial Function. Animals (Basel) 2019. 9. 10.3390/ani9050266.

Pi, D., Liu, Y., Shi, H., Li, S., Odle, J., Lin, X., Zhu, H., Chen, F., Hou, Y., and Leng, W. Dietary supplementation of aspartate enhances intestinal integrity and energy status in weanling piglets after lipopolysaccharide challenge. J Nutr Biochem 2014. 25, 456-462. 10.1016/j.jnutbio.2013.12.006.

Wang, X., Liu, Y., Li, S., Pi, D., Zhu, H., Hou, Y., Shi, H., and Leng, W. Asparagine attenuates intestinal injury, improves energy status and inhibits AMP-activated protein kinase signalling pathways in weaned piglets challenged with Escherichia coli lipopolysaccharide. Br. J. Nutr. 2015. 114, 553-565. 10.1017/s0007114515001877.

  1. Line 86: How were the animals divided? One piglet per box? Add details.

A: “In a temperature-controlled nursery barn (25 to 27°C), the piglets were kept separately in 1.80 × 1.10 m cages with unrestricted access to clean water and food” has been added in revised version. (Line 98-99)

  1. Line 112-114: Delete the sentence that is a duplication.

A: The duplicate sentences has been removed.

  1. Line 121: Specify the sampling site.

A: The middled ileum and jejunum was sampled, which was added in the revised version. (Line 131)

  1. Line 129-135: rewrite for clarity.

A: “Thaw the plasma samples out of the -80°C refrigerator and centrifuge at 3000 r/min for 10 min, then aspirate 150 μL of supernatant in a serum glass, which were detected using Automatic Biochemical Analyzer (Roche, Switzerland) and commercial kits (Sino-German Beijing Leadman Biotech Ltd, Beijing, China). And the indicators were detected include the following: Diamine oxidase (DAO), lactic acid (LACT). The concentrations of serum IL-1β, IL-6, IL-13, IL-4, TNF-α were measured with commercial ELISA kits (Cusabio Life Science Inc., Wuhan, China).” has been replaced by “Thaw the plasma samples out of the -80°C refrigerator and centrifuge for 10 min at 800 × g at 4°C, then aspirate 150 μL of supernatant in a serum glass, which were detected using Automatic Biochemical Analyzer (Roche, Switzerland) and commercial kits (Sino-German Beijing Leadman Biotech Ltd, Beijing, China). And the concentrations of blood indicators were detected include the following: diamine oxidase (DAO) and lactic acid (LACT).”. (Line 141-147)

  1. Line 137: Substitute with “Microtome sections of intestinal tissues were obtained from each sample and stained…” Add the thickness of the sections.

A: The revised sentences were presented and the thickness of the section has been added in Line 149-150.

  1. Line 144: Substitute with: “The grinded intestinal mucosa was analyzed”

A: The original sentences has been substituted with “The grinded intestinal mucosa was analyzed”. (Line 156)

  1. Line 169-177: substitute slice with sections

A: “slice” has been substituted with “sections”.

  1. Line 185 Statistical Analysis: Please add the experimental unit in the statistical evaluation.

A: The individual piglet used in this experiment served as the experimental unit for all data collection. (Line 213-214)

  1. Line 214: correct “trended” in “tended”

A: “trended” has been corrected in “tended”. (Line 241)

  1. Line 345: substitute” accumulating lines of literature” with “Many papers”

A: “accumulating lines of literature” has been substituted with “Many papers”. (Line 358)

  1. References: References in the text and in the references section are not in line with journal instruction for authors. Please revise this section.

A: Thanks for the reviewer’s suggestion. Citation of references in the text has been improved according to the Journal’s requirements.

Reviewer 3 Report

Authors Yu et al. submitted their manuscript entitled “Development of Intestinal Injury and Restoration of Weaned Piglets under Chronic Immune Stress to Antioxidants. The manuscript deals with an interesting topic and presents many results. Unfortunately, some parts in the manuscript were not possible to evaluate, e.g., due to the poor quality of some graphs and micrographs. It is a pity that the authors did not pay attention to the present manuscript to submit it in an acceptable form.

I miss some basic data dealing with LPS, e.g.,  DOI: 10.1002/chem.201403923 and 10.1016/j.molcel.2014.03.012, the influence of its structure/truncation on the virulence of G- bacteria in pigs DOI: 10.3390/toxins11090534 and DOI: 10.3390/toxins12090545, and a consequence of LPS release on organisms, e.g., DOI: 10.3389/fimmu.2014.00680. Moreover, there were monitored expressions of several inflammatory cytokines without mentioning their function and the meaning of the increase of the expression in individual cytokines.

There are my other objections, notices, and recommendations.

L97-104: Each of LPS treated and slaughtered piglets should have its control group with the same scheme as LPS injected but saline-treated.

 L114 and others: Elisa has to be written ELISA. Please, correct it through the manuscript. I will notice it more.

L116-118: This sentence repeated the sentence on lines 112-114. The first version is added O day. Please, correct the texts and truthfulness of the different information between these sentences.

L121-122: The location of samples should be more specified, e.g., proximal, middle, or distal jejunum. The jejunum is very long in pigs, e.g., compare to humans.

L125-126: Please, specify how did you take the intestinal mucosa.

L129: The information 3000 r/min does not allow us to imagine experimental conditions if the radius was not described. Please, use a centrifugation force including also temperature as it was done on line 117.

L134: IL-1b, IL-6, IL13, IL4, and TNF-a were selected. I did not find in the Introduction and Discussion any approvement of this selection. Moreover, I did not find any results dealing with your selected IL-13.

L145: Please, introduce SOD, GSH-Px abbreviations.

L155: Please, add the producer of Oligo 6.0 software.

L155: It would be more suitable to use at least two reference genes.

L156: Please, add amplification conditions and Real-Time PCR cycler information.

L169: I did not notice that the preparation of paraffin-embedded samples was described. Please, check it.

L172: Please, specify the primary antibody including its producer.

L183-184: Please, specify the Transmission Electron Microscope producer.

L189: It is necessary to take into account the differences between parametric and non-parametric tests. Please, realize that S.D. and S.E.M. are characteristics of parametric tests but median and modus are characteristics of the non-parametric test. It is impossible to use for comparison non-parametric tests and show mean+S.E.M as results. Please, rephrase the description dealing with statistics.

L193-194: In this text should be a curt description of results but not an explanation, justified, ... that belongs to the Discussion or Introduction. Please, remove this sentence.

L200-201: In this text should be a curt description of results but not an explanation, justification, ... that belongs to discussion or introduction. Please, remove this sentence. The described figures are of so poor quality that it is impossible to check if are correctly described. Please replace them with higher-quality figures.

L213: TJP - each used abbreviation has to be introduced.

Figure 2 (A) The micrographs are too small and without details that would allow reviewers to evaluate the differences. (B) Graphs and their descriptions are small and blurry. (C) The results are not presented as mean ± SEM, but as mean + SEM. Please, correct the description or graphs. The graphs and their description are a blur.

L228-229: In this text should be a curt description of results but not an explanation, justification, ... that belongs to discussion or introduction. Please, remove this sentence. The described figures are of so poor quality that is impossible to check if are correctly described. Please replace them with higher-quality figures.

L230: CD3+-positive proteins or cells?

L232: Proteins or cells?

L253: It could be the curt descriptions only.

L253: It could be the curt descriptions only.

L289-290: Here should be a curt description of results only.

Figure 6. The graphs are very small and blurry if they are increased. Please, modify the graphs to be readable.

L401-402: IL-6 can be produced also by other cells, e.g., enterocytes.

Hardly evaluate the description of the results if some of the micrographs are blurred and graphs not readable. It makes it impossible to properly evaluate the correctness of the following discussion.

Author Response

Reviewer: 3

I miss some basic data dealing with LPS, e.g., DOI: 10.1002/chem.201403923 and 10.1016/j.molcel.2014.03.012, the influence of its structure/truncation on the virulence of G-bacteria in pigs DOI: 10.3390/toxins11090534 and DOI: 10.3390/toxins12090545, and a consequence of LPS release on organisms, e.g., DOI: 10.3389/fimmu.2014.00680.

A: Thank you for your suggestion. LPS-induced chronic immune stress model has been built and reported by a previous study (Jia et al, 2015). Using this model, we found that pigs treated with LPS exhibited lower growth performance and muscle weight (Duan et al, 2019), further proving the validity of this model. In the present study, we aimed to investigate the effects of chronic immune stress on the development of intestinal injury and restoration of weaned piglets. Therefore, the information provided by the abovementioned references (e.g., the influence of its structure/truncation on the virulence of G-bacteria in pigs, and a consequence of LPS release on organisms) is not covered by this test.

Jia AF, Feng JH, Zhang MH, Li ZY, Chang Y, Xu SS, He C, Effects of chronic immunological stress induced by increasing administration of lipopolysaccharide on growth performance of piglets, Chin. J. Anim. Nutr., 2015, 27(6), 1868–1874.

Duan YH, Zheng CB, Zhong YZ, Song B, Yan ZM, Kong XF, Deng JP, Li FN, Yin YL. Beta-hydroxy beta-methyl butyrate decreases muscle protein degradation via increased Akt/FoxO3a signaling and mitochondrial biogenesis in weanling piglets after lipopolysaccharide challenge. Food & Function, 2019, 10: 5152.

Moreover, there were monitored expressions of several inflammatory cytokines without

mentioning their function and the meaning of the increase of the expression in individual cytokines.

A: Thank you for your suggestion. In the manuscript, we analyzed the expressions of TNF-α, IL-6, and IL-4, and their function was mentioned in Line 260. Among the three cytokines, we focused on IL-6, the detailed information was shown in Lines 416-422.

There are my other objections, notices, and recommendations.

  1. L97-104: Each of LPS treated and slaughtered piglets should have its control group with the same scheme as LPS injected but saline-treated.

A: Thank you for your suggestion. Firstly, our study focused on the development and modulation of intestinal injury and restoration in weaned piglets on longitudinal timeline. Moreover, published studies on the intestinal of weaned piglets have utilized Day as a repeated effect of LPS multi-injection or other stresses, which evaluated and elaborates the morphological and functional changes of intestine in response to these stresses and the associated regulatory mechanisms on a timeline.

Reference:

Escribano, D., Campos, P.H., Gutiérrez, A.M., Le Floc'h, N., Cerón, J.J., and Merlot, E. Effect of repeated administration of lipopolysaccharide on inflammatory and stress markers in saliva of growing pigs. Vet. J. 2014. 200, 393-397. 10.1016/j.tvjl.2014.04.007.

Hu, C.H., Xiao, K., Luan, Z.S., and Song, J. Early weaning increases intestinal permeability, alters expression of cytokine and tight junction proteins, and activates mitogen-activated protein kinases in pigs. J. Anim. Sci. 2013. 91, 1094-1101. 10.2527/jas.2012-5796.

Cao, S.T., Wang, C.C., Wu, H., Zhang, Q.H., Jiao, L.F., and Hu, C.H. Weaning disrupts intestinal antioxidant status, impairs intestinal barrier and mitochondrial function, and triggers mitophagy in piglets. J. Anim. Sci. 2018. 96, 1073-1083. 10.1093/jas/skx062.

  1. L114 and others: Elisa has to be written ELISA. Please, correct it through the manuscript. I will notice it more.

A: “Elisa” has been rewritten “ELISA” in revised version.

  1. L116-118: This sentence repeated the sentence on lines 112-114. The first version is added 0 day. Please, correct the texts and truthfulness of the different information between these sentences.

A: Thank for the reviewer’s suggestion. The duplicate sentences have been deleted and corrected in the revised version. (Line 126-128)

  1. L121-122: The location of samples should be more specified, e.g., proximal, middle, or distal jejunum. The jejunum is very long in pigs, e.g., compare to humans.

A: The middled ileum and jejunum was sampled, which was added in the revised version. (Line 131)

  1. L125-126: Please, specify how did you take the intestinal mucosa.

A: The specific steps for taking the intestinal mucosa has been described in revised version. (Line 135-139)

  1. L129: The information 3000 r/min does not allow us to imagine experimental conditions if the radius was not described. Please, use a centrifugation force including also temperature as it was done on line 117.

A: The centrifugal information has been corrected according to your suggestion, and the related information has been added in Line 143.

  1. L134: IL-1b, IL-6, IL13, IL4, and TNF-α were selected. I did not find in the Introduction and Discussion any approvement of this selection. Moreover, I did not find any results dealing with your selected IL-13.

A: Thanks for your careful checks. We are sorry for our carelessness. We did not intent to show the data IL-1b, IL-6, IL13, IL4, and TNF-α, which was not the results to be presented in this paper. We remove the description in revised version.

  1. L145: Please, introduce SOD, GSH-Px abbreviations.

A: The abbreviation has been introduced in revised version. (Line 157)

  1. L155: Please, add the producer of Oligo 6.0 software.

A: The producer of Oligo 6.0 software has been added in revised version. (Line 168)

  1. L155: It would be more suitable to use at least two reference genes.

A: Thanks for the reviewer’s suggestion. About 7 years ago, our lab carried out experiments according to the guideline described by Bustin et al (2009) to select appropriate reference genes for the intestine of pigs. The results showed that both GAPDH and β-actin are stably expressed in intestines of pigs. Both reference genes have been used in intestines of pigs by our labs (Han et al., 2021; Li et al., 20 19; Li et al., 2015; Luo et al., 2020; Ma et al., 2022; Ren et al., 2014). In this study, we used β-actin as a reference gene.

Reference: Han, H., Liu, Z., Yin, J., Gao, J., He, L., Wang, C., Hou, R., He, X., Wang, G., Li, T., et al. D-Galactose Induces Chronic Oxidative Stress and Alters Gut Microbiota in Weaned Piglets. Front Physiol 2021. 12, 634283. 10.3389/fphys.2021.634283.

Li, G., Li, J., Tan, B., Wang, J., Kong, X., Guan, G., Li, F., and Yin, Y. Characterization and Regulation of the Amino Acid Transporter SNAT2 in the Small Intestine of Piglets. PLoS. One. 2015. 10, e0128207. 10.1371/journal.pone.0128207.

Luo, W.J., Song, P., He, Z.M., Cao, S.P., Tang, J.Z., Xu, W.Q., Xiong, D., Qu, F.F., Zhao, D.F., Liu, Z., et al. JAK2 Mediates the Regulation of Pept1 Expression by Leptin in the Grass Carp (Ctenopharyngodon idella) Intestine. Front Physiol 2020. 11, 79. 10.3389/fphys.2020.00079.

Ma, J., Duan, Y., Li, R., Liang, X., Li, T., Huang, X., Yin, Y., and Yin, J. Gut microbial profiles and the role in lipid metabolism in Shaziling pigs. Anim. Nutr. 2022. 9, 345-356. 10.1016/j.aninu.2021.10.012.

Ren, W., Duan, J., Yin, J., Liu, G., Cao, Z., Xiong, X., Chen, S., Li, T., Yin, Y., Hou, Y., et al. Dietary L-glutamine supplementation modulates microbial community and activates innate immunity in the mouse intestine. Amino. Acids. 2014. 46, 2403-2413. 10.1007/s00726-014-1793-0.

The β-actin’s Ct values of all groups are provided in the following figure (Figure 1-2).

The statistical results of Ct values are shown in Figure 3-4.

Based on the data, we can know that β-actin was stably expressed in piglets’ intestine.

Figure 1 The Ct values of β-actin of jejunum

Figure 2 The Ct values of β-actin of ileum

Figure 3 The statistical results of Ct1and Ct2 values in jejunum

Figure 4 The statistical results of Ct1and Ct2 values in ileum

  1. L156: Please, add amplification conditions and Real-Time PCR cycler information.

A: thank for your suggestion and the relative information has been added in Line 169-173.

  1. L169: I did not notice that the preparation of paraffin-embedded samples was described. Please, check it.

A: The preparation of paraffin-embedded samples was added in Line 186-173.

  1. L172: Please, specify the primary antibody including its producer.

A: Thank for your suggestion, the primary antibody including its producer has been added in revised version. (Line 200-203)

  1. L183-184: Please, specify the Transmission Electron Microscope producer.

A: The Transmission Electron Microscope producer has been specified in the revised version.

  1. L189: It is necessary to take into account the differences between parametric and nonparametric tests. Please, realize that S.D. and S.E.M. are characteristics of parametric tests but median and modus are characteristics of the non-parametric test. It is impossible to use for comparison non-parametric tests and show mean+S.E.M as results. Please, rephrase the description dealing with statistics.

A: Thank for your suggestion. Whether the data satisfy normal distribution or homogeneity determined the tests we run in our studies. When the data with non-normal distributions or homogeneity, nonparametric Kruskal-Wallis test was used. And the parametric test was used when meeting the conditions above. The published paper showed that the non-parametric test was used when the data did not meet the conditions above, but the results are still presented as means ±S.E.M. or S.D (Liu et al., 2021; Xiao et al., 2021; Zhao et al., 2021). We rephased the description dealing with statistics. (Line 215-218)

References:

Liu, Y., Li, Y.J., Loh, Y.W., Singer, J., Zhu, W., Macia, L., Mackay, C.R., Wang, W., Chadban, S.J., and Wu, H. Fiber Derived Microbial Metabolites Prevent Acute Kidney Injury Through G-Protein Coupled Receptors and HDAC Inhibition. Front. Cell. Dev. Biol. 2021. 9, 648639. 10.3389/fcell.2021.648639.

Xiao, Z.P., Lv, T., Hou, P.P., Manaenko, A., Liu, Y., Jin, Y., Gao, L., Jia, F., Tian, Y., Li, P., et al. Sirtuin 5-Mediated Lysine Desuccinylation Protects Mitochondrial Metabolism Following Subarachnoid Hemorrhage in Mice. Stroke 2021. 52, 4043-4053. 10.1161/strokeaha.121.034850.

Zhao, X., Liu, Y., Ding, H., Huang, P., Yin, Y., Deng, J., and Kong, X. Effects of Different Dietary Protein Levels on the Growth Performance, Serum Biochemical Parameters, Fecal Nitrogen, and Carcass Traits of Huanjiang Mini-Pigs. Front Vet Sci 2021. 8, 777671. 10.3389/fvets.2021.777671.

  1. L193-194: In this text should be a curt description of results but not an explanation, justified, ... that belongs to the Discussion or Introduction. Please, remove this sentence.

A: The sentence has been removed in revised version.

  1. L200-201: In this text should be a curt description of results but not an explanation, justification, ... that belongs to discussion or introduction. Please, remove this sentence. The described figures are of so poor quality that it is impossible to check if are correctly described. Please replace them with higher-quality figures.

A: The sentence has been removed in revised version. The described figures have been replaced with higher-quality figures.

  1. L213: TJP - each used abbreviation has to be introduced.

Figure 2 (A) The micrographs are too small and without details that would allow reviewers to evaluate the differences. (B) Graphs and their descriptions are small and blurry. (C) The results are not presented as mean ± SEM, but as mean + SEM. Please, correct the description or graphs. The graphs and their description are a blur.

A: The abbreviation has been introduced in revised version. According to your suggestion, the graphs and their description has been corrected in revised version.

  1. L228-229: In this text should be a curt description of results but not an explanation, justification, ... that belongs to discussion or introduction. Please, remove this sentence. The described figures are of so poor quality that is impossible to check if are correctly

described. Please replace them with higher-quality figures.

A: The sentence has been removed in revised version. The described figures have been replaced with higher-quality figures.

  1. L230: CD3+-positive proteins or cells?

A: Thank for the reminder, the CD3+-positive proteins has been corrected to CD3+-positive cells.

  1. L232: Proteins or cells?

A: Thank for the reminder, the proteins has been corrected to cells.

  1. L253: It could be the curt descriptions only.

A: The explanatory sentence has been removed in revised version.

  1. L289-290: Here should be a curt description of results only.

2Figure 6. The graphs are very small and blurry if they are increased. Please, modify the graphs to be readable.

A: The explanatory sentence has been removed in revised version. According to your suggestion, the graphs has been modified in revised version. And the Figure 6 has been modified in revised version.

  1. L401-402: IL-6 can be produced also by other cells, e.g., enterocytes.

A: Thank for the suggestion, which has been added in the revised version. (Line 415)

  1. Hardly evaluate the description of the results if some of the micrographs are blurred and graphs not readable. It makes it impossible to properly evaluate the correctness of the following discussion.

A: Thanks for your correction. In our resubmitted manuscript, the micrographs and graphs has been revised.

Round 2

Reviewer 3 Report

The authors submitted a revised version of their manuscript that was highly improved. The main objection to the revised version is the inappropriate use of statistics.

Parametric tests: VALUES are compared, and characteristics are mean, S.D. and S.E.M

Nonparametric tests: ORDERS of values are compared, and the characteristics are median, modus, and ranges.

I know that it is possible to find nonparametric tests and presentation results as mean, S.D. or S.E.M in highly respected journals. However, it doesn't mean that it is correct.

There is a part of the text of the Author's instruction of the American Association of Microbiologists for their highly respected journals:

https://journals.asm.org/doi/epub/10.1128/IAI.00811-13:

"Keep in mind that nonparametric tests do not compare the means of the data. For this reason, data that are analyzed using nonparametric tests should be reported as medians and ranges, or medians and selected percentiles (typically 25th and 75th percentiles), instead of means and standard deviations. "

Thus, I repeatedly conclude that results obtained with nonparametric tests would not be presented as mean and S.E.M.

Author Response

We are very grateful to Reviewer for reviewing the paper so carefully. It is really true as Reviewer suggested that results obtained with nonparametric tests would not be presented as mean and S.E.M. We presented the results obtained with nonparametric tests in the boxplot or column graph as median with interquartile range (Figure 2 B,C; Figure 3 A,B; Figure 6 ) . And median values with capital letters were considered to be significantly different (P < 0.05). (Line 214-220, 250-255, 277-280, 354-356)